# Noise Pollution and Its Correlations with Occupational Noise-Induced Hearing Loss in Cement Plants in Vietnam

**DOI:** 10.3390/ijerph18084229

**Published:** 2021-04-16

**Authors:** Tinh Thai, Petr Kučera, Ales Bernatik

**Affiliations:** 1Faculty of Safety Engineering, VŠB-Technical University of Ostrava, 70030 Ostrava, Czech Republic; ales.bernatik@vsb.cz; 2Department of Fire Protection, VŠB-Technical University of Ostrava, 70030 Ostrava, Czech Republic; petr.kucera@vsb.cz

**Keywords:** cement manufacturing, noise-induced hearing loss, noise level

## Abstract

Noise-Induced Hearing Loss (NIHL) is a global issue that is caused by many factors. The purpose of this study was to survey noise level to identify NIHL and its relationship with other factors in cement plants in Vietnam. Noise level was measured at one cement plant and three cement grinding stations located in the South of Vietnam. The audiometric data of exposed workers were surveyed to determine NIHL. Finally, the relationship between NIHL and noise level in cement plants was determined. The results show that noise level in almost all processes exceeded the permissible exposure limit (PEL). In this study, 42 cases (10% of exposed workers) with occupational NIHL were found with mean age (SD) of 49 (9.0) years. All NIHL cases were found in the departments in which the noise level exceeded the PEL, which included quarry (*n* = 16), maintenance (*n* = 12), production (*n* = 10), co-waste processing (*n* = 3) and quality assurance (*n* = 1). There was a positive and significant correlation between the NIHL and the excessive noise exposure in the cement plants (r = 0.89, *p* = 0.04).

## 1. Introduction

Noise-induced hearing loss (NIHL) has long been recognized as an occupational disease, amongst copper workers from hammering on metal, blacksmiths in the 18th century, and shipbuilders or boilermakers after the Industrial Revolution [1]. It has been suggested that 12% or more of the global population is at risk of hearing loss from noise [2]. The World Health Organization estimated that one-third of all cases of hearing loss can be attributed to noise exposure [3]. In cement plants, noise is typically one of the most harmful factors and is generated in many processes. Noise sources in a cement plant are varied and mainly include gas dynamic noise, which is induced by blower, air compressor and dust collector operation; mechanical noise, which is induced by machine operation, such as crushers, mills; electric–magnetic noise, which is induced by electric motors. The noise levels in some production areas, such as in the cement mill, air compressor, blower or crusher, in the cement grinding station ranged between 89 and 105 dBA and exceeded the Vietnamese permissible exposure limit (PEL) (85 dBA/08 working hours) [4]. Workers who are under long-term exposure to excessive noise level can develop noise-induced hearing loss (NIHL) [5].

In Vietnam, occupational NIHL is recognized as a work-related health condition and is compensated. According to Vietnamese legislation, the noise level in the workplace in all businesses is surveyed at least once a year, and workers who operate in areas which exceed the permissible exposure limit (PEL) undertake a yearly audiometric examination to determine whether there is occupational NIHL. However, a full hearing-conservation program has not yet been required by law, nor has it been adequately implemented by cement makers. The lack of awareness of NIHL among workers, employers and even health-care professionals leads to the high prevalence of NIHL observed in Asian laborers [5]. In order to understand the correlations between noise level and the risk of NIHL in the cement industry in Vietnam, the purpose of this study was to (1) survey the noise level in the main processes of one cement plant and three cement grinding stations; (2) identify NIHL in cement plants; (3) analyze the correlations between NIHL and the noise level in cement manufacturing plants.

## 2. Materials and Methods

This study was conducted in November 2018 at one cement plant (S1) and three cement grinding stations (S2, S3, S4) located in the South of Vietnam. The general information of study sites is described in Table 1.

Firstly, the noise level was measured for 30 cement manufacturing sub-processes at the study sites in November 2018. At each measurement site, the noise level was measured in the workers’ hearing zone for three minutes during each of 5 different working hours (7:30; 9:30; 11:30; 13:30; 15:30). These hours cover the duration of the day shift and represent the noise levels for all workers at that site, since operational processes are the same for all shifts. Noise level was measured during normal full operating conditions without interference from other factors, such as rain or traffic activities, and was generated continuously during measurements. Noise level was measured using a sound-level meter (Quest SoundPro SE, Oconomowoc, WI, USA), placed at the worker’s working position and at a height of 1.4 m from the ground. Before measuring, the noise meter was calibrated to the Vietnamese legal requirement. For each measurement site, the equivalent noise level was calculated from the five measurements using the formula (1) [6]:(1)$$\mathrm{Leq}=10log\left(\frac{1}{n}\left[10^{0.1L_{1}}+10^{0.1L_{2}}+\ldots +10^{0.1L_{n}}\right]\right)$$
where Leq: A-weighted, is the equivalent steady sound level of noise energy averaged over time; *L_n_*: the *n*th sound level of *n* sound levels (*n* = 5).

Secondly, the authors worked with the health practitioners who are in charge of occupational health checks at the plants. In 2018, 418 workers (60.67% of excessive noise-exposed workers) at the study sites were sent to the Institute of Public Health, Ho Chi Minh City for NIHL test. The general processes to identify the occupational NIHL, applied in Vietnam, included (1) exposed workers were initially evaluated the auditory threshold at 4000 Hz; (2) workers with a reduction threshold of 50 dB or greater at 4000 Hz were further tested at the frequency 500 Hz, 1000 Hz, 2000 Hz and 8000 Hz; (3) the percentage hearing loss (PHL) was identified by using the Fowler–Sabine table [7]. Then, the percentage of impairment loss was compared to WHO NIHL criteria to define occupational NIHL. The grades of hearing impairment included slight or mild impairment with a hearing loss range from 26 to 40 dB; moderate impairment with a hearing loss range from 41 to 60 dB; severe impairment with hearing loss range from 61 to 80 dB; profound impairment with a hearing loss range of 81 dB or greater [8].

Thirdly, the authors worked with Human Resources Departments at study sites to survey the age, working department, and length of service of occupational NIHL cases. Finally, the correlation between occupational NIHL and the noise level in the stages of the cement-making process was analyzed.

## 3. Results and Discussion

### 3.1. Excessive Noise Level and Wide Range Distribution in Cement Plant

The results of the noise level surveillance in one cement plant and three cement grinding stations reveal that the equivalent noise level in many processes of cement manufacturing exceeded the permissible exposure limit for an 8-hour shift, regulated by the Vietnamese Standard (85 dBA). The processes with an excessive noise level at the first study site (S1) included quarry, coal mill, cement mill, kiln, air compressor, and co-waste-processing area. The processes with an excessive noise level at the second study site (S2) included cement ball mill, air compressor room, and packer. The processes with an excessive noise level at the third study site (S3) was the vertical cement mill, and at the fourth study site (S4) this included the vertical cement mill and compressor room. The noise levels in the main processes of cement manufacturing plant ranged between 68.8 dBA and 103.3 dBA, which workers were exposed to every single day. The noise levels at the study sites are shown in Table 2.

In a full cement manufacturing plant (S1), the equivalent noise level in almost all processes exceeded the permissible exposure limit (PEL) regulated by the Vietnamese standard or by the National Institute for Occupational Safety and Health (NIOSH) (85 dBA for 08 working hours). At the quarry, the first process in cement manufacturing, the equivalent noise level in all sub-processes, including drilling, bulldozer operation, limestone crusher, and crusher hopper, exceeded the PEL and ranged between 86.0 dBA and 101.5 dBA. The clinker production process with two main sub-processes included raw material grinder and rotary kiln. The equivalent noise level at raw material grinder was 91.2 dBA and exceeded the PEL by 6.2 dBA, and at the kiln, the noise level was 90.1 dBA, which exceeded the PEL by 5.1 dBA. In the parallel process, the equivalent noise level at the coal mill was detected as the highest noise level, at 88.3 dBA, which is 3.3 dBA above the PEL. At the cement production process, the equivalent noise level at the cement mill was 90.3 dBA (+5.3 dBA) and in co-waste processing, the equivalent noise level at the waste cutter was 90.8 dBA, which exceeded the PEL. The equivalent noise level at the quality assurance process was 83.2 dBA, but at times, the noise level was 85.5 dBA.

At the three cement grinding plants (S2, S3, S4), the equivalent noise level ranged between 68.8 dBA and 103.3 dBA. The highest equivalent noise level in these processes at S2 was 93.2 dBA; at S3 it was 89.0 dBA; at S4 it was 103.32 dBA. The noise source in cement manufacturing varies with a high level and a wide distribution.

The above results reveal that the noise level in the main processes of cement plants exceeded the PEL, and these results are comparable to other studies. Previous studies showed that the processes in the cement plant that exceeded the PEL included quarry (98 dBA), crusher room (106 dBA), ball mill room (101 dBA), raw material mill (103 dBA), rotary kiln (93 dBA), coal mill (102 dBA), and air compressor room (95 dBA) [8,9,10,11]. The equivalent noise level in the cement industry is comparable to the noise levels in other industries, such as a textile mill in Mainland China (84–103 dBA), transport in India (89–106 dBA), agriculture in Japan (82–99 dBA), and oil refinery in Taiwan (73–89 dBA) [5].

Noise is typically one of the most hazardous factors in cement manufacturing. It has the potential to lead to occupational deafness [10], and workers who are daily expose to such excessive noise levels for a prolonged period will face considerable social and physiological impacts, including NIHL [12].

### 3.2. Noise-Induced Hearing Loss in Cement Plant

A total of forty-two occupational NIHL cases have been recognized for the period of operations from 1996 to 2018 in the study sites. The number of the occupational NIHL cases found at study sites from 1996 to 2018 is shown in Table 3. The characteristics of NIHL and non NIHL cases are shown in Table 4.

The first case of occupational NIHL was detected in 2003 at the cement plant (S1) after six years of operation. During the first ten years of operation at S1, three occupational NIHL cases were found and, within next five years, through 2011, an additional fifteen occupational NIHL cases had been recorded. By the end of 2015, after 18 years of operation, forty-one occupational NIHL cases were found at S1 and one case was recorded at S2. There have been no cases of occupational NIHL detected at S3 or S4 so far. Almost all occupational NIHL cases were found at S1 (*n* = 41), which had been in operation for 21 years, and one case was found at S2. For employees with NIHL (*n* = 42), the mean duration of their employment when NIHL detected was 12 years. The first case of NIHL was found after 2 years of employment at S1.

The mean (SD) age of excessive noise exposed workers surveyed in 2018 (*n* = 418) was 38.6 (9.0) years; the youngest workers were 22 years old (*n* = 8) and the oldest workers were 59 years old (*n* = 4). The mean (SD) age of NIHL cases was 49 (9.0) years and mean (SD) age of the non NIHL cases was 37.4 (8.5) years. The mean age of NIHL cases found at the study sites was higher than the mean age of the audiometric participants in 2018 and of the non-NIHL cases. In this study, the youngest worker that suffered from NIHL was 36 years old and the oldest worker was 57 years old. This finding was comparable with the study among metallurgical company workers in Brazil, which reported that the mean age of those with NIHL was 50 (12) years [13] and the mean age of participants at six quarries in Malaysia with NIHL was 35 (9) years [9].

The percentages of NIHL cases with age groups from 41 to 50 years old and above 50 years old were 45% (*n* = 19) and 52% (*n* = 22), respectively, which was higher than that for the same age group of non-NIHL workers.

In this study, the mean (SD) length of working service of all audiometric participants surveyed in 2018 was 12.6 (6.2) years. The mean (SD) length of working service of non-NIHL cases was 11.8 (6.1) years. The mean (SD) length of working service of NIHL cases was 21 (3.5) years, which was higher than that of the non-NIHL cases.

### 3.3. The Correlation between the Occupational NIHL and the Excessive Noise Level

In this research, all of the occupational NIHL cases were discovered at departments where the equivalent noise level exceeded PEL (85 dBA/8 hours). In total, 16 occupational NIHL cases (38%) were discovered at the quarry, where the equivalent noise level ranged between 86.0 dBA and 101.5 dBA; 12 cases (29%) were discovered in the maintenance department, and 10 cases (24%) in the production department where the equivalent noise level ranged between 88.3 dBA and 96.9 dBA; 3 cases (7%) came from the co-waste processing department, where the equivalent noise level was 90.8 dBA, and 1 case (2%) came from the quality assurance department, where the equivalent noise level was 83.2 dBA; however, sometimes during operation the noise level exceeded 85 dBA. 

The correlation between the number of NIHL cases in different departments and the excessive noise level in their departments was significant (r = 0.89, *p* = 0.04). The relationship between NIHL and prolonged high-noise exposure was published in other studies. A study in Ghana showed that the noise level in corn mills exceeded the PEL (85 dBA, WHO, 1999) and 23% of workers in corn mills presented with evidence of NIHL [14]. This finding was also comparable to those reported in other studies [15,16,17].

### 3.4. The Degree of NIHL in Cement Plants

Almost all workers in cement plants had been working in environments with a high noise level. In 2018, 74% workers (*n* = 393) at S1, 45% workers (*n* = 88) at S2, 59% workers (*n* = 95) at S3 and 28% workers (*n* = 113) at S4 were often exposed to the high noise level in their daily lives. Through this exposure, and improper NIHL-prevention programs, workers in cement manufacturing plants are at a high risk of NIHL.

The degree of hearing loss in this study ranged from mild to severe, based on the WHO classification [8]. In this study, of 42 workers with NIHL, 30 cases (71.4%) had mild hearing loss, 10 cases (23.8%) had moderate hearing loss and 2 cases (4.8%) had severe hearing loss. A study of NIHL among quarry workers in the North-Eastern State of Malaysia showed that 42% (*n* = 23) had mild and moderate hearing loss and 16% (*n* = 9) had severe hearing loss [9]. In addition, a study of NIHL in cement plants in Nigeria reported that 67% of workers had mild sensorineural hearing loss and 12% had moderate hearing loss in the right ear [18].

### 3.5. Hearing Conservation Program

The hearing conservation program (HCP) comprises a set of coordinated measures to prevent the development of occupational NIHL [18]. The HCP is an essential service that employers are required to implement in many developed countries, such as The United States of America, Canada, Argentina. The effectiveness of an HCP has been studied and reported in many studies [19,20,21]. A specific HCP needs to be considered and addressed by the cement makers in Vietnam to diminish NIHL. The elements of the HCP included noise hazard identification and noise exposure monitoring; controlling noise exposure; hearing protection devices; audiometric testing; hazard communication; education and training; recordkeeping; continuous monitoring and improvement.

The limitations of this study were that data concerning general health conditions and whether hearing-protection devices were habitually used during daily work activities were not recorded. These factors could affect the incidence of hearing loss of workers in cement manufacturing plants.

## 4. Conclusions

Noise is one of the main occupational health hazards in the main processes of cement manufacturing plants. Long-term exposure to excessive noise pressure levels exceeding the PEL (85 dBA) within cement manufacturing processes could lead to NIHL. There was a positive correlation between the number of cases of NIHL and excessive noise exposure in cement manufacturing processes. The degree of NIHL in cement plants ranged between mild and severe loss of hearing. In addition, the prevalence and variety of noise distribution in the day-to-day exposure of workers in cement plants could account for the increase in the number of NIHL cases in the cement plant. It is necessary for cement makers and health and safety personnel in cement plants in Vietnam to enforce the hearing conservation program to reduce the risks of NIHL for workers.

## Figures and Tables

**Table 1 ijerph-18-04229-t001:** The General Information of Study Sites.

Site	Established Year	Capacity (Mt/Year)	Full Time Equivalent in 2018	Noise Exposed-Workers in 2018	Operation Type
S1	1997	3.6	393	290	Cement plant
S2	2003	0.5	88	40	Cement grinding station
S3	2005	3.0	95	56	Cement grinding station
S4	1996	1.5	113	32	Cement grinding station

**Table 2 ijerph-18-04229-t002:** The Equivalent Noise Level at the Cement Plants.

Study Sites	Process	Area	Equivalent Noise Level (dBA)					Leq (dBA)
			7:30	9:30	11:30	13:30	15:30	
S1	Quarry	Drilling machine	90.1	88.3	87.6	91.2	91.2	89.9
		Cab of bulldozer	87.7	88.2	83.1	82.4	85.6	86.0
		Crusher of limestone	93.4	95.3	97.4	99.1	99.0	97.4
		Crusher hopper	99.2	100.1	101.2	102.4	103.5	101.5
	Clinker and cement production	Coal mill area	89.3	83.2	87.8	88.3	90.2	88.3
		Compressor room	98.2	98.3	94.5	95.9	96.4	96.9
		Raw mill	89.7	90.8	91.3	91.6	92.3	91.2
		Kiln area	88.6	87.9	89.3	92.1	91.1	90.1
		Cement mill	88.6	90.3	87.9	91.2	92.1	90.3
	Quality Assurance	Lab room	82.3	80.3	81.5	84.2	85.5	83.2
	Co-waste processing	Waste cutter and feeding chute	88.3	92.5	88.6	92.4	90.2	90.8
S2	Unloading material	Unloading	72.1	72.4	73.5	76.1	73.5	73.8
	Storage	Storage	67.8	66.3	68.5	72.1	66.8	68.8
	Pre-grinding	Roller press	83.5	82.3	82.4	84.5	86.3	84.1
	Grinding	Ball mill	95.2	92.1	90.0	94.3	92.7	93.2
		Air compressor	89.3	91.4	88.4	91.3	88.2	89.9
	Dispatch	Packer	86.2	84.3	85.3	86.7	83.4	85.3
		Truck loader	72.3	72.5	74.5	76.8	77.5	75.2
S3	Unloading material	Jetty	71.3	77.8	73.3	75.4	77.5	75.7
	Storage	Warehouse	76.3	75.4	77.2	75.5	78.9	76.9
	Raw material feeding	1st floor	77.8	83.5	80.8	81.3	83.2	81.7
	Grinding	Vertical mill	84.7	86.2	90.4	92.0	87.8	89.0
	Dispatch	Packer	80.8	85.3	81.2	80.7	80.2	82.1
		Truck loader	75.3	72.6	73.2	74.9	76.5	74.7
S4	Raw material preparation	Pozzolana crusher	82.3	81.2	84.3	84.3	81.5	82.9
	Storage	Pozzolana storage	72.1	75.2	76.4	74.3	75.2	74.9
	Grinding	Vertical mill	88.1	89.6	89.3	88.7	89.3	89.0
		Compressor	93.2	103.6	105.4	104.3	103.2	103.3
	Dispatch	Packer	81.2	80.4	83.2	82.3	81.2	81.8
		Truck loader	77.4	80.2	79.2	78.4	75.2	78.4

**Table 3 ijerph-18-04229-t003:** The Number of the Occupational NIHL Cases Found at Study Sites from 1996 to 2018.

Site	2003	2004	2005	2006	2007	2008	2009	2010	2011	2012	2013	2014	2015	2016	2017	2018
S1	1	2	0	0	1	0	3	0	11	7	1	5	10	0	0	0
S2	0	0	0	0	0	0	0	0	0	0	0	1	0	0	0	0
S3	0	0	0	0	0	0	0	0	0	0	0	0	0	0	0	0
S4	0	0	0	0	0	0	0	0	0	0	0	0	0	0	0	0

**Table 4 ijerph-18-04229-t004:** The Characteristics of NIHL and Non-NIHL Cases at Study Sites.

Variable		Audiometric Participants (*n* = 418)	NIHL Cases (*n* = 42)	Non-NIHL Cases (*n* = 376)
		Mean (SD)	Mean (SD)	Mean (SD)
Age (years)		38.6 (9.0)	49 (9.0)	37.4 (8.5)
Gender				
Male		402	42	360
Female		16	0	16
Age group (years)				
20 years old and below		0	0	0
21 to 30 years old		77	0	77
31 to 40 years old		181	1	180
41 to 50 years old		106	19	87
51 years old and above		54	22	32
Length of working service (years)		12.6 (6.2)	21 (3.5)	11.8 (6.1)
Hearing loss degree				
20–40 dB		NA	30	NA
41–60 dB		NA	10	NA
61–80 dB		NA	2	NA
Hearing loss level (dB)				
Left ear	500 Hz	NA	35.4 (11.1)	NA
	1000 Hz	NA	39.6 (9.5)	NA
	2000 Hz	NA	37.9 (9.9)	NA
	4000 Hz	NA	53.9 (7.7)	NA
Right ear	500 Hz	NA	35.3 (10.7)	NA
	1000 Hz	NA	38.6 (10.6)	NA
	2000 Hz	NA	38.4 (9.5)	NA
	4000 Hz	NA	54.6 (8.0)	NA
Percentage hearing loss (%)				
Left ear		NA	35.8 (14.7)	NA
Right ear		NA	35.5 (15.0)	NA

NA: Not applicable.

## Data Availability

Data from this study are available upon request.

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
