# Peer review of "Noise Pollution and Its Correlations with Occupational Noise-Induced Hearing Loss in Cement Plants in Vietnam"

_ijerph, 2021, doi:10.3390/ijerph18084229_

Round 1
Reviewer 1 Report
The authors study the effect of noise in manufacturing plants where workers are exposed to a huge excess of noise allowed by present regulations. The study is of primary importance and the data presented show the need for new working conditions to diminish the hearing injuries in the plants.
I suggest the publication of the paper after some minor corrections are performed, and the questions carefully considered:
(1) (170-171) table --> Table.
(2) (173) Table 4: for which frequencies the loss occurred or was measured?
General questions:
(1) Can one compare the hearing loss with normally aging people at the same age reported in Table 4? Is there a significant difference?
(2) What are the possible remedies for these working persons to diminish
the effect of high noise?
Author Response
Dear Sir/Madam,
We thank you so much for your time and your valuable comments. We have reviewed your comments, have addressed them as follow and revised in the manuscript accordingly. We would like to resubmit the manuscript for publication, so please let us know what we need to do to proceed further.
Thank you
Point 1: (170-171) table --> Table
Response 1: This point has been corrected, on the line 151-152 of the revised manuscript.
Point 2: Table 4: for which frequencies the loss occurred or was measured?
Response 2: The more frequency characteristics of the NIHL have been provided in Table 4 on the line 154. The audiometric was measured at 500 Hz, 1000 Hz, 2000 Hz, and 4000 Hz. The hearing loss level in dB was most reduced at frequency of 4000 Hz. The mean hearing loss level (dB) of left ear at 4000 Hz was 53.9 (7.7) and the mean hearing loss level (dB) of right ear at 4000 Hz was 54.6 (8.0). The percentage hearing loss (%) of left ear was 35.8 (14.7) and of right ear was 35.5 (15.0).
Point 3: General questions: Can one compare the hearing loss with normally aging people at the same age reported in Table 4? Is there a significant difference?
Response 3: We thank the reviewer for the interesting suggestion of comparison of the hearing loss with the normally aging people. Unfortunately, we did not add this comparison into the manuscript adequately because of information shortage. The mean age of NIHL in this study was 49 (9.0) years. This finding was comparable with the study among metallurgical company workers in Brazil, which reported that the mean age of those with NIHL was 50 (12) years and the mean age of participants at six quarries in Malaysia with NIHL was 35 (9) years. (on the line 168-175).
Point 4: What are the possible remedies for these working persons to diminish
the effect of high noise?
Response 4: In order to diminish the effects of high noise exposure, we suggest that a hearing conservation program (HCP) should be implemented by cement makers. We added a little bit of detail of the hearing conservation program in the lines 213-222. The hearing conservation program is a set of coordinated measures to prevent the development of occupational NIHL [18]. The HCP is a must duty that requires employer to implement in many developed countries such as The United States of America, Canada, Argentina. The effectiveness of an HCP has been studied and reported in many researches [18-20]. A specific HCP need to be considered and addressed by the cement makers in Vietnam to diminish the NIHL. The elements of an HCP included noise hazard identification and noise exposure monitoring; controlling noise exposure; hearing protection devices; audiometric testing; hazard communication, education and training; recordkeeping, and continuous monitoring and improvement.

Reviewer 2 Report
- The article documents the correlation between excessive exposure to noise and the likelihood of noise induced hearing loss (NIHL) in the cement-production facilities that are studied. This could make it a positive and useful contribution. However, there are many problems. Perhaps the most significant is extremely poor use of English. Normally, I would provide editing comments and suggestions, but the problems are too numerous and too extreme in this case so I have to suggest that the authors get some professional help for this.
- Another problem is acknowledged by the authors themselve in the paragraph on lines 231-233. This paragraph has the only mention of use of hearing-protective devies in the whole paper. Could use of ear defenders have prevented all of the NIHL? Was there any use of them at all? The final sentence of the paper states "It is necessary for cement workers, health and safety personnel in cement plants in Viet Nam to enforce the hearing conservation program to reduce the risk of NIHL for workers." A little bit of detail about what the "hearing conservation program" would include would greatly enhance the paper.
- It might also be useful to provide a bit of detail about frequency characterisitcs of the NIHL. Was it a sloping hearing loss? Was it more clearly characterized by a loss at any particular frequency?
- I believe that for IJERPH the Abstract should be a single paragraph without subheadings and it should be limited to about 200 words. The submitted abstract needs to change for both reasons.
Author Response
Dear Sir/Madam,
We thank you so much for your time and your valuable comments. We have reviewed your comments, have addressed them as follow and revised in the manuscript accordingly. We would like to resubmit the manuscript for publication, so please let us know what we need to do to proceed further.
Thank you
Point 1: The article documents the correlation between excessive exposure to noise and the likelihood of noise induced hearing loss (NIHL) in the cement-production facilities that are studied. This could make it a positive and useful contribution. However, there are many problems. Perhaps the most significant is extremely poor use of English. Normally, I would provide editing comments and suggestions, but the problems are too numerous and too extreme in this case so I have to suggest that the authors get some professional help for this.
Response 1: The manuscript has undergone English language editing by MDPI. The text has been checked for correct use of grammar and common technical terms.
Point 2: Another problem is acknowledged by the authors themselves in the paragraph on lines 231-233. This paragraph has the only mention of use of hearing-protective devices in the whole paper. Could use of ear defenders have prevented all of the NIHL? Was there any use of them at all? The final sentence of the paper states "It is necessary for cement workers, health and safety personnel in cement plants in Viet Nam to enforce the hearing conservation program to reduce the risk of NIHL for workers." A little bit of detail about what the "hearing conservation program" would include would greatly enhance the paper.
Response 2: The paragraph of the previous version on the lines 231-233 is our description of the limitations of the research. The NIHL is caused by many factors such as excessive noise level exposure, ageing, genetic and individual behaviors. In this study, we have just focused on the relationship between the NIHL and the excessive noise level in cement manufacturing processes. We agree that use of ear defenders such as earmuff or earplug is the last measure in hierarchy of risk control strategy to prevent the NIHL. A hearing conservation program mentioned in the following can be used to reduce the risk of NIHL.
We added a little bit of detail of the hearing conservation program in the lines 213-222. The hearing conservation program (HCP) is a set of coordinated measures to prevent the development of occupational NIHL [18]. The HCP is a must duty that requires employer to implement in many developed countries such as The United States of America, Canada, Argentina. The effectiveness of an HCP has been studied and reported in many researches [18-20]. A specific HCP need to be considered and addressed by the cement makers in Vietnam to diminish the NIHL. The elements of an HCP included noise hazard identification and noise exposure monitoring; controlling noise exposure; hearing protection devices; audiometric testing; hazard communication, education and training; recordkeeping, and continuous monitoring and improvement.
Point 3: It might also be useful to provide a bit of detail about frequency characteristics of the NIHL. Was it a sloping hearing loss? Was it more clearly characterized by a loss at any particular frequency?
Response 3: The frequency characteristics of the NIHL have been provided in Table 4 on the line 154. The hearing loss level in dB was mostly reduced at the frequency of 4000 Hz. The mean hearing loss level (dB) of left ear at 4000 Hz was 53.9 (7.7) and the mean hearing loss level (dB) of right ear at 4000 Hz was 54.6 (8.0). The percentage hearing loss (%) of left ear was 35.8 (14.7) and of right ear was 35.5 (15.0).
Point 4: I believe that for IJERPH the Abstract should be a single paragraph without subheadings and it should be limited to about 200 words. The submitted abstract needs to change for both reasons.
Response 4: We revised the abstract following your comment, the abstract is combined in one paragraph and limited to about 200 words.

Round 2
Reviewer 2 Report
The authors responded appropriately to my comments. They added some information that I requested and received help with English language usage from editors at MDPI. While this editing has resulted in improvements, there are still many problems and I have compiled a long list of suggested edits. I have uploaded these suggestions in a Word file

Author Response
Dear Reviewer,
We thank you for your thorough review and comments. We revised all point-by-point as your suggestions. Please see the attachment for the detail.
Thank you
The authors
